# PointARSim: Point-cloud–enhanced Generative Auto-Regressive Simulation for Closed-loop End-to-end Autonomous Driving Evaluation

## Abstract

End-to-end autonomous driving (E2E-AD) has garnered increasing attention in recent years. However, exist evaluation fails to simultaneously meet the requirements for both interactivity (e.g., CARLA) and realism (e.g., nuScenes). In this work, we introduce *PointARSim*, an autoregressive generative simulation framework that leverages point cloud skeleton representation accumulated from offline datasets as conditions for diffusion models to synthesize multi-view images. Specifically, we separately aggregate foreground-background point cloud, facilitating utilization of visual geometry. To enable accurate control of the generated driving scene, we design a multi-conditioning ControlNet that integrates point cloud priors, previous-frame condition and optional backgrounds injection, guided by a geometry-aware cycle loss. Furthermore, to assess generative models' fidelity when the ego-vehicle's trajectory differs greatly from recorded offline logs, we introduce a new evaluation protocol based on the nuPlan dataset, called *nuPlan-SimGenEval* benchmark. Extensive experiments on both nuScenes and nuPlan-SimGenEval demonstrate that the proposed approach significantly outperforms existing methods, highlighting its potential for next-generation closed-loop autonomous driving simulation.

## 1 Introduction

End-to-end autonomous driving (E2E-AD) (Li et al., 2024; Yang et al., 2025; Jia et al., 2025) has attracted increasing attention in recent years. However, benchmarking of E2E-AD algorithms remains a challenge for the community. Specifically, existing benchmarks can be broadly divided into two categories: 1) **Open-loop** benchmarks like nuScenes (Caesar et al., 2020) fail to provide dynamic sensor feedback based on the decisions from the E2E-AD models, resulting in biased assessment (Dauner et al., 2023). 2) Conversely, **close-loop** simulations like CARLA (Dosovitskiy et al., 2017; Zhang et al., 2023a) offer a reactive system. Nevertheless, notable gaps remain between game engines and real world from both rendering and behavioral perspectives.

A promising direction is to use generative models (Ho et al., 2020) to generate realistic multi-view images based on novel simulator trajectories. Note that unlike street-scene video generation (Wang et al., 2024b; Yang et al., 2024a) or reconstruction (Yan et al., 2024b), as shown in Fig. 1, generative simulators must satisfy the following requirements: (I) **Instantaneous interactivity**: the future of the world depends on the decision of the E2E-AD model at each time step and thus enforces an autoregressive process, which prevents application of the video generation models. (II) **Views from arbitrary location**: the evaluated E2E-AD model may have arbitrary trajectories and thus requires the simulator to generate images from diverse viewpoints. Directly applying reconstruction methods lead to severe view inconsistency due and result in collapse (Wang et al., 2024a). (III) **ID consistency of dynamic agents**: the simulator need to ensure consistency of surrounding agents across frames and views, requiring precise control over each agent's motion.

Although existing works have made great progress in city scene generation, few methods are capable of meeting all the above requirements simultaneously. These approaches either rely on video generation (Wang et al., 2024a; Yan et al., 2024a; Guo et al., 2025) with future trajectory or struggle to

achieve specific agent control (Yang et al., 2024a;b). More crucially, most existing work relies on the nuScenes (Caesar et al., 2020) for benchmark, which does not allow for customized agent trajectories and thus fails to assess the generative simulator's performance under challenging conditions with significant drift from the recorded offline logs (Zhai et al., 2023).

In this work, we delve into the design of such an autoregressive generative simulator, exploring three key perspectives: log data utilization, model design, and benchmarking. (I) **For offline log data**, we develop a Point Cloud Skeleton system that not only uses offline data as labels for training of the generative model, but also **maintains a representation of the scenario as inputs during autoregressive generation**. Specifically, we decouple and accumulate the scene-level point cloud into background and foreground assets, and then edit point clusters based on simulate signal of current time step. The instantaneously updated Point Cloud Skeleton contains visual and geometry information of the scene to guide the generation. (II) **For the design of the generator**, we integrate previous frame information and depth/color priors derived from the Point Cloud Skeleton with ControlNet (Zhang et al., 2023b), with an optional background insert for potentially key traffic signals. Concretely, we introduce Point Cloud Skeleton-guided data augmentation, noisy previous condition, and Geometry-Aware Cycle loss to **balance conflicts among various conditioning signals** and ultimately enhance the fidelity. (III) **For generative simulator benchmarking**, we propose an evaluation protocol for generative simulation using the nuPlan (Caesar et al., 2021) dataset, called **nuPlan-SimGenEval**. There are two tracks: *log-replay*, which is similar to the classic setting, and *Interactive Divergence*, which makes the ego vehicle execute a vastly different trajectory compared to original logs.

Empirically, our method outperforms existing approaches in both nuScenes and nuPlan dataset. Under the perception evaluation with *Interactive Divergence*—the setting closest to realistic simulation—our method achieves ∼7% improvement over the current SOTA. Demonstrating the effectiveness of the proposed *PointARSim* techniques.

## 2 RELATED WORKS

### 2.1 URBAN-SCENE GENERATIVE MODELS

Conditional diffusion models have recently demonstrated remarkable success in visual content generation (Rombach et al., 2022), particularly due to their ability to generate high-quality images from abstract conditions. With the expansion of conditioning mechanisms, ranging from text and semantic maps to bounding boxes and maps, they become applicable for autonomous driving scenarios (Yang et al., 2023b; Gao et al., 2023).

Recently, research investigates video diffusion models as predictive world models (Yang et al., 2024a; Gao et al., 2024b; Wang et al., 2024b), aiming to build differentiable and generalizable simulators. These models attempt to rollout future pixel states based on abstract actions or text inputs. However, they often fall short in maintaining coherent world dynamics and typically lack interactive or feedback-responsive capabilities, which is required for realistic simulation. To this end, DriveArena (Yang et al., 2024b) incorporates traffic simulator into the system, enabling reactive and temporally extended control. However, conditioning directly on previously generated frames undermines the auto-regressive process. During simulation, where past frames are model outputs rather than ground-truth, recursive conditioning amplifies small generative errors over time, ultimately leading to severe output degradation or collapse.

### 2.2 NOVEL VIEW SYNTHESIS

3D reconstruction offers an alternative to synthesizing high-fidelity novel-view images from recorded data. Methods based on NeRF (Mildenhall et al., 2021) and 3D Gaussian Splatting (3DGS) (Kerbl et al., 2023) have shown impressive performance in generating sensor data along recorded trajectories in urban scenes (Yang et al., 2023a; Zhou et al., 2024; Xu et al., 2024). However, as pointed out by FreeVS (Wang et al., 2024a), **classical reconstruction methods face a limitation of data convergence when synthesizing from significantly shifted viewpoints.** To overcome this, FreeVS (Wang et al., 2024a) introduces a strategy that project neighbor-frame point clouds as conditions, however, this approach fails in interactive simulations because the representation from adjacent frames cannot support large trajectory deviations from the original logged data. FreeSim (Fan et al., 2024) further extends novel view generation by gradually applying spatial shifts, enabling this process with only image data.

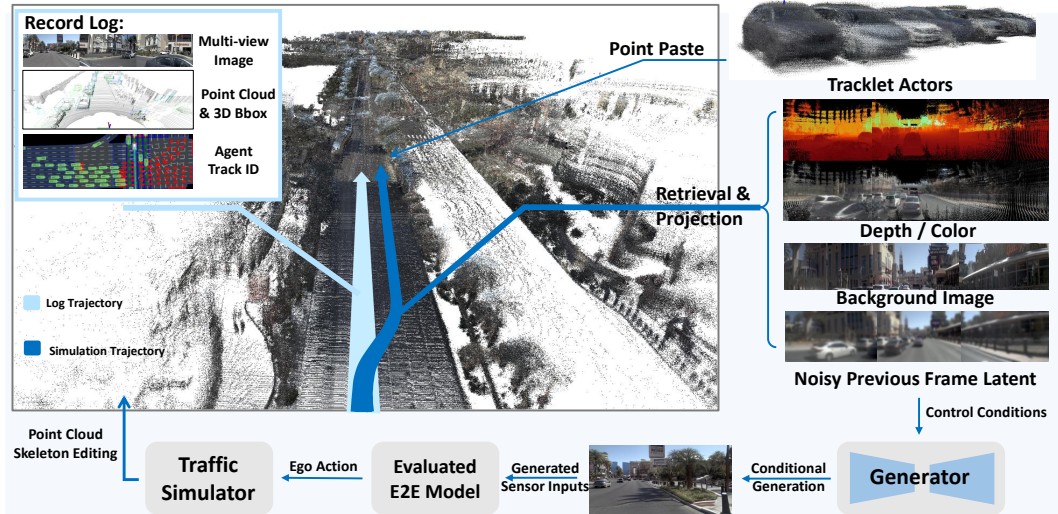

Figure 1: *PointARSim* **Generative Simulator.** We utilize offline logs to build the decoupled foreground-background Point Cloud Skeleton. At each time step, according to the current action of the evaluated E2E-AD model, the traffic simulator rollouts the future position of ego and other vehicles. Based on the updated actor information, the Point Cloud Skeleton is updated and then projected back to camera view as multiple conditions. Finally, the Generator generates the latest sensor images in the new location, which is fed into the E2E-AD model for next-step action.

Nevertheless, these reconstruction-generation hybrid method are primarily designed for pre-recorded static scenes and lack the ability to dynamically control each drivable agent. In addition, current NVS evaluation protocols typically focus on lateral (Y-axis) viewpoint shifts while ignoring ego-vehicle rotations (Z-axis yaw), limiting their applicability to generative simulators that require full 6-DoF control for realistic driving simulations.

## 3 METHOD

**Overview.** To fulfill the need for both step-by-step interaction in simulation tasks and consistency in long-form video generation, we employ an autoregressive framework aid by editable point geometry. As illustrated in Fig. 1, our generative simulator is designed to produce images in an interactive loop with the E2E-AD model, and a Point Cloud Skeleton system is used as geometric guidance for generating each frame. Further details are discussed below.

### 3.1 POINT CLOUD SKELETON

The motivation of building the Point Cloud Skeleton comes from the observation that **the simulation process is based on offline logs.** Thus, the detail and geometry of background are static and could be obtained from scene-level point cloud accumulation over frames, while the geometry of foreground (e.g., vehicles) could be obtained by accumulating points within the 3D bounding at different frames. Both information could aid the temporal consistence of autoregressive generation process.

In this work, we introduce a color-depth dual stream point cloud aggregation strategy, which allows the traffic simulator to directly control the movement of foreground points within the background scene, and provide camera-view conditions from arbitrary viewpoints. As illustrated in Fig. 2, we first project each frame's point cloud and image pixels into ego-centric coordinate system to obtain colorized point cloud. Then, based on detection and tracking annotations, we segment the colorized points into dynamic $P_t^{\text{inbox}}$ and static $P_t^{\text{outbox}}$ in frame $t$.

$$P_t^c = P_t^{\text{inbox}} \cup P_t^{\text{outbox}}, \quad P_t^{\text{inbox}} \cap P_t^{\text{outbox}} = \emptyset \tag{1}$$

The segmented background points are then aggregated along the recorded trajectory with $N$ frame in the global coordinate system using transfer matrix $T_{\text{global}}(t)$, while the actor points is aggregated in actor-local coordinate using $T_{\text{actor}}(t)$, according to track id $i$.

$$P_{\text{global}}^{\text{background}} = \bigcup_{t=1}^{N} T_{\text{global}}(t) \cdot P_t^{\text{outbox}}, P_{\text{actor}(i)} = \bigcup_{t=1}^{N} T_{\text{actor}(i)}(t) \cdot P_t^{\text{inbox}(i)} \tag{2}$$

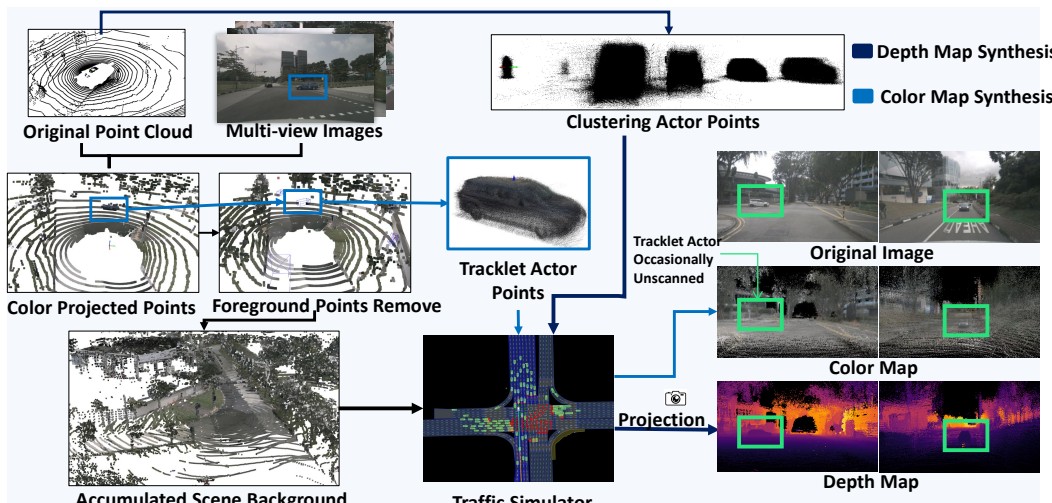

Figure 2: **Construction and Utilization of Point Cloud Skeleton.** We separately accumulate foreground and background skeleton with point cloud and images from recorded logs. During generation, the accumulated foregrounds instances is pasted into the background according to the simulator control. This skeleton provides two current conditioning inputs for the generation: the color map, which with rich image details but lacks novel-view coherence for moving agents, and the depth map which offers robust geometric consistency across different viewpoints.

To ensure efficient storage and robust rendering from arbitrary viewpoints, background point clouds are voxel-downsampled. All assets are then organized by city and tracking ID for convenient retrieval. During the simulation process, the foreground is moved within the background point cloud according to the movement of the bounding boxes in the traffic simulator. Finally, the Point Cloud Skeleton is projected onto each camera's image plane to synthesize the color map used as the condition for the diffusion generation model.

However, the only color map suffers from a critical limitation, as shown in Fig. 2 bottom-right: **most agents are not fully scanned, which leads to significant incompletion color map when the ego-vehicle deviates from the recorded trajectory during simulation**, as shown in Fig. 5-FreeVS. This inconsistency introduces notable distributional gaps between autoregressive model training and evaluation, demonstrates that only-color representation are ill-suited for simulation scenarios.

We devise a depth map branch to mitigate the aforementioned issue. Given the large spatial scale of outdoor scenes (∼100m), depth maps are relatively insensitive to fine variations in agent appearance (∼0.5m). Leveraging this property, we cluster annotated bounding boxes by category and size to construct dense point cloud templates for common traffic participants, such as sedan, SUV, and pedestrian, as in Fig. 2 (top-right). These dense templates are then combined with the background point cloud to synthesize high-quality depth maps. It enhances the consistency of geometric conditions **across diverse simulation viewpoints, especially for partially scanned agents.**

### 3.2 GENERATOR DESIGN

For sensor data generation, we use Controlnet structure (Zhang et al., 2023b) to autoregressively generate images based on conditions from the current simulator state, as shown in Fig. 3. Given the diverse and potentially conflicting conditions, the generator is optimized to balance two objectives: employing its learned prior to generate dynamic foregrounds while leveraging low-level cues from the colorized point to enforce visual fidelity and temporal coherence. To address these, we introduce three modules: Color Actor Vibration, Noise Previous Latent, and Geometry-Aware Cycle Loss.

**Color Actor Vibration.** As discussed in Sec. 3.1, the offline point cloud representation suffers from a limitation: during interactive simulation, where the ego-vehicle deviates from the logged trajectory, **novel instances with sparse point cloud coverage frequently appear, creating a mismatch with the dense data used during training**,as shown in Fig. 5 - FreeVS, compromising the generation. To address this, we provide a consistent geometric prior by introducing dense depth templates for actors. However, naively training the model to fuse these conditions is insufficient to elicit the desired generative behavior. Specifically,the generative process is a diffusion model $\theta$, which predicts noise $\epsilon$

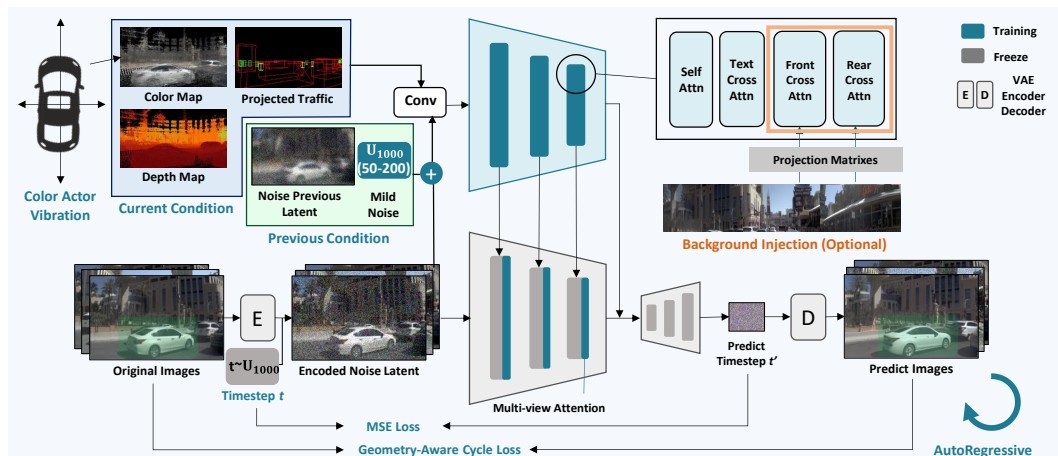

Figure 3: **Generator Pipeline.** Autoregressive training are prone to overfitting when conditioning inputs, such as projected point clouds and past frames, closely resemble the training target. To mitigate this, we design to selectively extract the most accurate elements from diverse conditioning inputs while filtering out irrelevant or conflicting information. We also introduce an optional background injection module designed to enhance details in far-field traffic signs, detailed in the Appendix.

from a noisy latent $z_t$, conditioned on a color map $C_{color}$, a depth map $C_{depth}$ and previous frame $C_{temp}$. The standard training objective is:

$$\mathcal{L}(\theta) = \mathbb{E}_{(I,C_{color},C_{depth},C_{temp},\epsilon)} \left[ \|\epsilon - \epsilon_\theta(z_t, t, C_{color}, C_{depth}, C_{temp})\|^2 \right] \qquad (3)$$

a trivial solution exists due to the high correlation between the training condition $C_{color}$ and the ground-truth image $I$. The model can achieve low loss by learning a degenerate function that primarily relies on $C_{color}$ while ignoring $C_{depth}$. This leads to fails when incomplete actor appear from novel trajectory during inference.

To counteract this, we introduce Color Actor Vibration (CAV) as a shown in Fig. 4. During training, each colorized actor added to the background is perturbed by a random displacement sampled from a normal distribution $\mathcal{N}(0,1)$ along the x and y axes. Since the depth map $C_{depth}$ remains geometrically robust, this forces $\epsilon_\theta$ to become a non-trivial solution, learning to use $C_{depth}$ for geometric consistency while treating the $C'_{color}$ as auxiliary guidance.

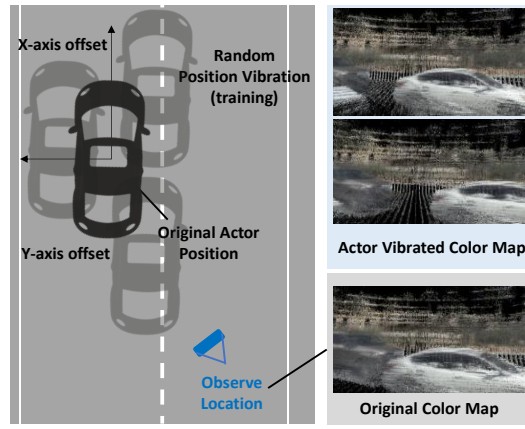

Figure 4: **Color Vibration Data Augment.**

**Noise Previous Latent.** By nature, the simulator restricts the use of future trajectories, requiring the model conditions on the previous frame, $C_{temp}$. However, the high similarity between adjacent frames in driving scenarios leaves a shortcut in Eq. 4: despite degenerating $C_{color}$, $C_{temp}$ remains a strong proxy for the target $I$, allowing the model to learn to simply copy this prior. During autoregressive inference, this shortcut causes minor generative errors to propagate and accumulate, leading to a gradual degradation in quality and eventual collapse, as shown in Fig. 5-Drive Arena.

To mitigate this, we introduce a **unified degeneration strategy** in both training and testing. We formulate the temporal prior, $C'_{temporal}$, by injecting mild noise into the previous latent, $z_{t-1}$, over a few forward diffusion steps $((t' \in [50, 200]))$. ensuring to preserve the high-level semantic content of the prior while corrupting the low-level textures that enable the reconstructive shortcut. Since $t'$ is independent of the primary diffusion timestep $t$, the model is encouraged to disentangle the reference content from the target generation. Finally, our objective is rebuild as:

$$\mathcal{L}(\theta) = \mathbb{E}_{(I,C_{depth},\epsilon)} \left[ \|\epsilon - \epsilon_\theta(z_t, t, C'_{color}, C_{depth}, C'_{temporal})\|^2 \right] \qquad (4)$$

**Sensitive to Colored Point Cloud**    **Accumulate Error of Autoregressive Generation**

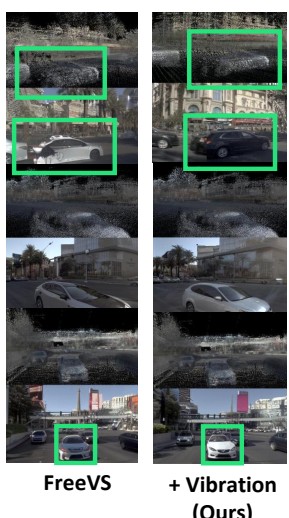 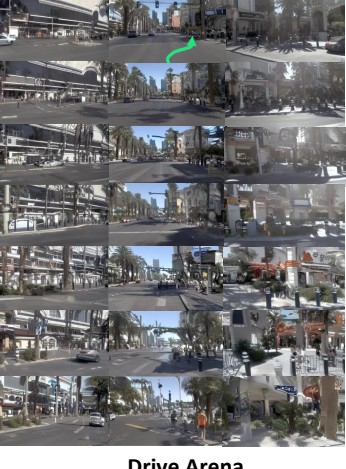 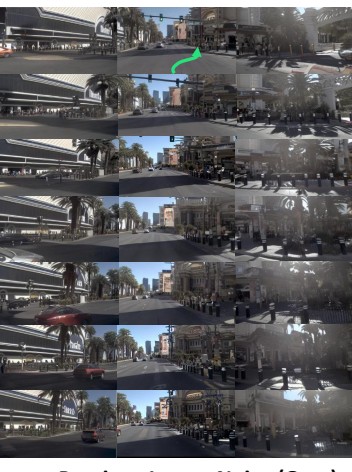

**FreeVS**    **+ Vibration (Ours)**    **Drive Arena**    **+ Previous Latent Noise (Ours)**

Figure 5: **Issues Observed in Interactive Divergence Test**. Simple project color points as FreeVS (Wang et al., 2024a) is sensitive to the completion and sparsity of color actors point in autoregressive situation. Drive Arena (Yang et al., 2024b) accumulates error during autoregressive generation while the proposed previous latent noise strategy significantly improves it.

compelling the model to synthesize novel content instead of merely replicating reference pixels.

**Geometry-Aware Cycle Loss.** Following above process, $C'_{color}$ can be considered the target $I$ with perturbed actor positions. Meanwhile, if the model only predicts the intrinsic noise steps of $C'_{temporal}$, it can also recover a target with perturbed actor. To prevent the model from learning an incorrect "recovery" pattern, we introduce an actor position loss to strengthen the conditional weight of the depth map. However, a loss designed to reinforce instance positions is ineffective when applied directly in the latent domain (Fig. 6). We therefore opt for a cycle-style reinforcement strategy. Specifically, since applying a full cycle loss during diffusion training is computationally expensive, for a predicted timestep $t'$ during training, we have:

$$\hat{x}_0 = D_{vae}\left(\frac{1}{\sqrt{\bar{\alpha}_{t'}}}\left(z_{t'} - \sqrt{1-\bar{\alpha}_{t'}}\,\epsilon_\theta(z_{t'}, t')\right)\right) \quad (5)$$

$$\mathcal{L}_{\mathcal{ID}} = Mask_{obj}\left(Mask_t\,\|\hat{x}_0 - x_0\|_2^2\right) \quad (6)$$

where $\hat{x}_0$ denotes the one-step sampled image from the noisy latent $z_{t'}$, and $Mask_t$ selects batches with low noise levels ($t<200$), $Mask_{obj}$ isolates depth-instance aligned regions using the projected bounding boxes. Finally ,we add $\mathcal{L}_{\mathcal{ID}}$ to the original diffusion loss to jointly optimize the training objective.

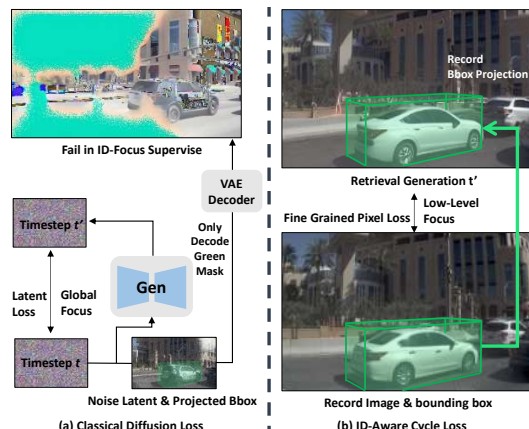

Figure 6: **Geometry-Aware Cycle Loss.** Directly apply weight loss in latent space is ineffective.

### 3.3 NUPLAN-SIMGENEVAL

Generative simulation aims to synthesize sensor data along **arbitrary trajectories** within fixed street environments. However, existing methods typically restrict testing to the original recorded trajectories (Caesar et al., 2020), which could not fully reflect the ability of simulation. As a result, there are several overlooked issues among existing methods, as illustrated in Fig. 5.

To this end, we introduce **nuPlan-SimGenEval**, an evaluation track for models of generetive simulation based on nuPlan (Caesar et al., 2021). It is composed of two tracks as shown in Fig. 7: (a) The **Log-Replay** track samples one frame every 4 frames for testing, with the remainder for training,

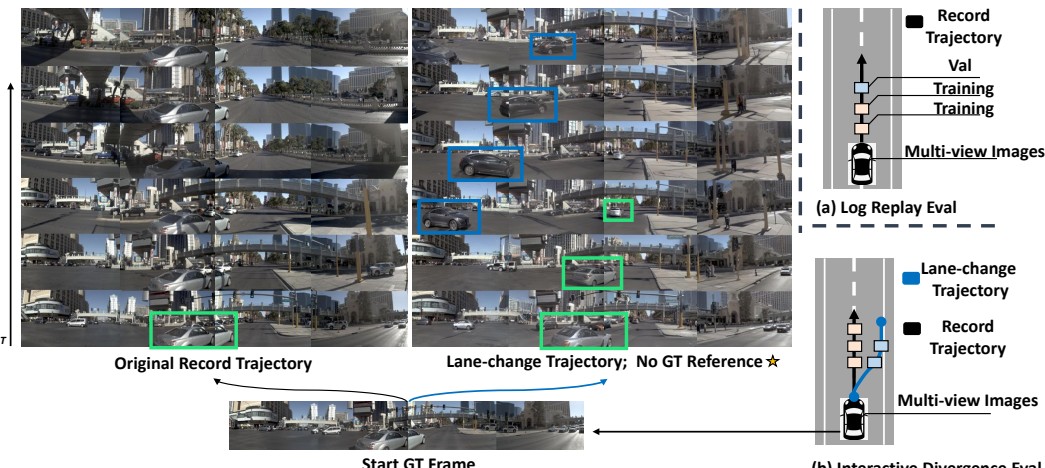

Figure 7: **nuPlan-SimGenEval.** Starting with GT frame, (a) Log-Replay assesses the generation performance on the original trajectory; (b) Interactive Divergence assesses the generation performance when inter-actor dynamics deviate from the original log.

following previous sets (Zhou et al., 2024; Yan et al., 2024b; Yang et al., 2023a). (b) The novel **Interactive Divergence** track applies a manually designed trajectory for the ego vehicle, which is significantly divergent from the original trajectory, while training is still on the original path. In contrast to prior evaluation protocols, the predefined trajectory lacks ground-truth reference, making long-horizon generation more challenging and better aligned with generative simulation.

For evaluation, we employ both the FID score and perception tasks. The perception tasks leverage the property of the Point Cloud Skeleton: by marking movable object points and projecting them onto the image plane, we obtain segmentation labels under arbitrary viewpoints, as shown in Fig. 8. Based on this, we advocate evaluating perception robustness using semantic segmentation models.

## 4 EXPERIMENTS

### 4.1 DATASET & IMPLEMENT DETAILS

We conduct experiments on the nuScenes and nuPlan-mini (nuPlan-SimGenEval) datasets. For nuScenes, we use 700 scenes for training and 150 scenes for validation. Evaluation is performed using standard metrics, including FID, FVD, and perception scores assessed with UniAD (Hu et al., 2023). For nuPlan-mini, we use the first sensor-blob data ($\sim 22000 \times 8$ frames) due to limited computation, which is comparable to nuScenes. For Log-Replay track, we report FID and FVD scores, while for the more challenging Interactive Divergence track, we integrate the nuPlan simulator, and additionally report Mask2Former (Cheng et al., 2022) vehicle segmentation scores.

For fair comparison with existing models (Gao et al., 2023; Yang et al., 2024b), we base our generative model on SDv1.5 (Rombach et al., 2021), generating images with a resolution of $224 \times 400$. The training process involves 200 epochs with a batch size of 16 on the nuScenes dataset, and 150 epochs with a batch size of 8 on the nuPlan dataset. For the nuScenes experiments, we use ASAP (Wang et al., 2023) to interpolate 12Hz annotations and use bicubic interpolation to upsample image for the driving agent. For more details and hyper-parameters regarding point cloud skeleton, generator, and Interactive Divergence track, see Appendix.

### 4.2 MAIN RESULTS

#### 4.2.1 QUANTITATIVE ANALYSIS

**nuScenes.** We first evaluate the image synthesis performance of *PointARSim* on the nuScenes validation dataset. As shown in Table 1-nuScenes, our method achieves state-of-the-art performance in terms of generation quality, achieving significantly lower FID and FVD scores. Furthermore, we evaluate UniAD on the nuScenes dataset. As shown in Table 2, UniAD demonstrates improved planning performance on image sequences generated by us, while maintaining competitive segmentation and detection accuracy under the standard nuScenes open-loop evaluation protocol.

Figure 8: **Point Skeleton provides segmentation GT for the evaluation of Interactive Divergence**.

Table 1: **Comparison of Log-Replay Generation.** For nuScenes FVD evaluation, we test with 17 frame, following previous works (Wen et al., 2024b), and use 30 frames for the nuPlan-mini test.

| Method | Conditon | Resolution | nuScenes FID$_{(\downarrow)}$ | nuScenes FVD$_{(\downarrow)}$ | nuPlan Log-Replay FID$_{(\downarrow)}$ | nuPlan Log-Replay FVD$_{(\downarrow)}$ |
|---|---|---|---|---|---|---|
| Panacea (Wen et al., 2024b) | Layout+Text | 256×512 | 16.96 | 139.00 | - | - |
| Panacea+ (Wen et al., 2024a) | Layout+Text | 256×512 | 15.50 | 103.00 | - | - |
| MagicDrive (Gao et al., 2023) | Layout+Text | 224×400 | 16.20 | 217.94 | - | - |
| MagicDiT (Gao et al., 2024a) | Layout+Text | 224×400 | 20.91 | 97.21 | - | - |
| DriveArena (Yan et al., 2024a) | Layout+Text+Prev | 224×400 | 16.03 | 185.32 | 11.52 | 123.69 |
| Ours | Layout+Point+Prev | 224×400 | **7.09** | **69.47** | **8.87** | **100.34** |

Table 2: **UniAD's Different Tasks in nuScenes Log-Replay Generation.**

| Source Data | 3DOD ↑ mAP | 3DOD ↑ NDS | Segmentation ↑ Lanes | Segmentation ↑ Drivable | Segmentation ↑ Divider | Segmentation ↑ Crossing | L2 ↓ 1.0s | L2 ↓ 2.0s | L2 ↓ 3.0s |
|---|---|---|---|---|---|---|---|---|---|
| ori nuScenes (Caesar et al., 2020) | 37.98 | 49.85 | 31.31 | 69.14 | 25.93 | 14.36 | 0.51 | 0.98 | 1.65 |
| MagicDrive (Gao et al., 2023) | 12.92 | 28.36 | 21.95 | 51.46 | 17.10 | 6.57 | 0.57 | 1.14 | 1.95 |
| Panacea (Wen et al., 2024b) | 13.72 | 27.73 | 18.23 | 52.37 | 17.21 | 6.32 | 0.58 | 1.14 | 1.97 |
| DriveArena (Yang et al., 2024b) | **16.06** | 30.03 | 26.14 | **59.37** | 20.79 | 8.92 | 0.56 | 1.10 | 1.89 |
| Ours | 15.82 | **32.87** | **26.37** | 57.66 | **21.59** | **9.18** | **0.53** | **1.06** | **1.83** |

**nuPlan-SimGenEval.** The nuPlan dataset has more diverse and challenging evaluations. Table 1 (Right) gives the results on Log-Replay track. The proposed method outperforms layout-only based approach by producing more realistic images with improved long-range temporal stability. Table 3 gives results on Interactive Divergence track. We compare FID between the new and original trajectories and the perception performance. Due to the substantial differences between trajectories, FID becomes less indicative. Still, the proposed approach achieves state-of-the-art results across all metrics, with a notable improvement of 7% in the perceptual test.

### 4.2.2 QUALITATIVE ANALYSIS

**Unseen Trajectory Generation.** As illustrated in Fig. 7 & 5, *PointARSim* is able to synthesize long-horizon, high-fidelity, and identity-consistent multi-view camera sequences, especially **for generative simulation: in Fig. 7, the ego-vehicle drift onto sidewalks.**

**Failure Cases.** Due to projection, all three conditions can fail when a vehicle is extremely close to the ego-car. As shown in Fig 9, **color and depth maps become incomplete or distorted, and the projected layout cannot keep full boxes at such proximity.** Consequently, generated actors may re-enter with **shape distortion** and **color inconsistency** after disappearing. **This failure also occurs in layout-only models.** Moreover, such near-field relations are absent in our dataset, leaving this regime under-supervised.

Table 3: **Results on the Interactive Divergence Track.** *We autoregressive reproduce the design of this method with aligned settings e.g. resolution and backbone.

| Method | Condition | Design Mode | FID$_{(\downarrow)}$ | IoU(%) |
|---|---|---|---|---|
| MagicDrive (Gao et al., 2023) | Layout+Text | Image | 47.15 | 42.47 |
| DriveArena (Yang et al., 2024b) | Layout+Text+Prev | Autoregressive Image | 44.88 | 46.89 |
| FreeVS* (Wang et al., 2024a) | Points+Layout+Prev | Video | 27.82 | 54.52 |
| Ours | Points+Layout+Prev | Autoregressive Image | **27.73** | **61.31** |

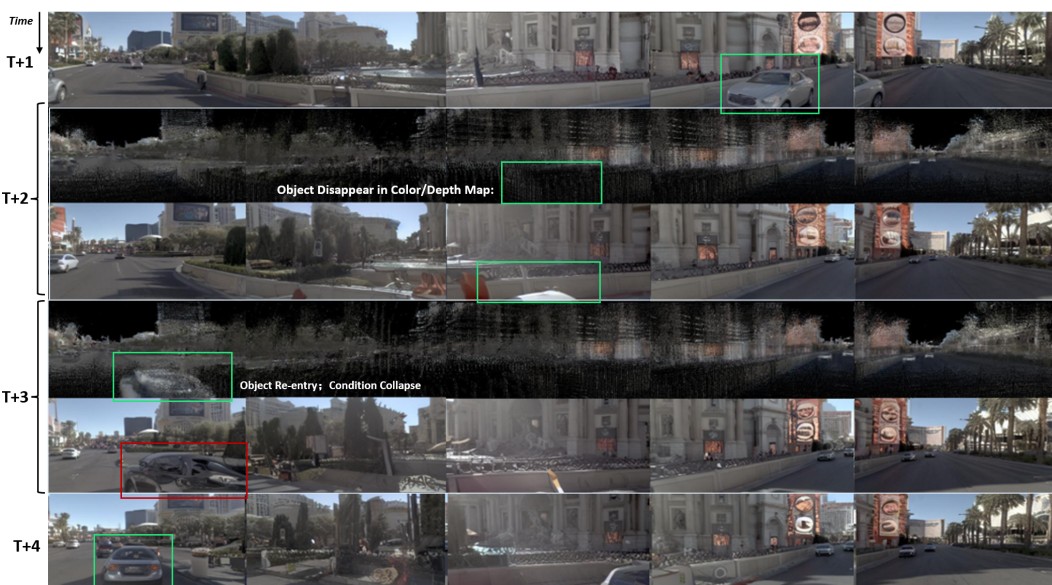

Figure 9: Failure cases caused by inaccurate bounding boxes and point cloud skeleton.

Table 4: **Ablation Study.** "LO" denotes projected layout; "(N)Prev" refers to (Noisy) Previous Latent; "Clr" and "Depth" represent point conditions; "CAV" stands for Color Actor Vibration. FID$_{I,S}$ indicate scores under interpolation and simulation tests. **Segmentation scores are most indicative of generation quality in simulation scenarios.**

| Exp. | Condition | FID$_{I(\downarrow)}$ | FVD$_{I(\downarrow)}$ | FID$_{S(\downarrow)}$ | Seg (IoU) |
|---|---|---|---|---|---|
| 1 | LO+Prev | 11.52 | 123.69 | 44.88 | 46.89 |
| 2 | LO+Prev+Clr | **8.62** | **98.43** | 38.27 | 50.91 |
| 3 | LO+(N)Prev+Clr | 9.22 | 104.95 | 27.82 | 54.52 |
| 4 | LO+(N)Prev+Clr+Depth | 9.89 | 107.23 | 28.45 | 55.67 |
| 5 | LO+(N)Prev+Clr+Depth+CAV | 8.92 | 103.45 | 28.26 | 58.87 |
| 6 | LO+(N)Prev+Clr+Depth+CAV+Cycle | 8.87 | 100.34 | **27.73** | **61.31** |

### 4.3 ABLATION STUDY

As illustrated in the Fig. 5, using only previous frames or the color map independently leads to a collapse in generation quality. Therefore, in Table 4 Exp. 1 and 2, we introduce both the point cloud and previous-frame conditions simultaneously. However, we observe that providing multiple strong conditions can **cause the model to overfit to the most dominant cue**, reducing generalization. Table 4 Exp. 2-4 support this observation. When we intentionally degrade both the color map and previous-frame inputs, the model achieves significantly better performance, indicating improved robustness. Furthermore, introducing a cycle consistency loss further enhances the overall model quality, as shown in Table 4 Exp. 5-6.

### 5 CONCLUSION

In this work, we incorporate point cloud data to investigate the construction of generative simulators from the perspectives of data processing, model design, and evaluation. Our study highlights the significant potential of generative models in advancing autonomous driving simulation, particularly

in enhancing realism and diversity. Furthermore, we provide an in-depth analysis of multi-conditional generation in road scenarios, offering insights into the complementary strengths and limitations of different conditioning inputs. Extensive experiments across multiple datasets demonstrate that our approach consistently outperforms existing methods, establishing a new benchmark in the community.

**ETHICS STATEMENT.** Our research aims to advance the field of autonomous driving and does not present immediate, direct negative societal impacts. The primary goal of this work is to provide a potential simulation environment for E2E-AD.

All datasets used in this study, namely nuScenes, and nuPlan, are publicly available and have been widely adopted by the machine learning community for academic research. These datasets consist of image, point cloud and do not contain personally identifiable information or sensitive content.

We acknowledge that the advancement of Generative Model carries a potential for misuse in broader applications. However, our work is foundational in nature and does not develop a deployable application. We have focused on the theoretical and empirical aspects of the proposed method. We encourage future research building upon our work to consider the specific ethical implications of their target applications. We believe that the potential benefits of this research in fostering more efficient and capable machine learning systems outweigh the foreseeable risks.

**REPRODUCIBILITY STATEMENT.** To ensure the reproducibility of our results, we provide a comprehensive description of our methodology, implementation details, and experimental setup in the main paper. Furthermore, we commit to making our resources publicly available. The source code for our experiments, implemented in Python using the PyTorch framework, will be released on GitHub upon publication of this paper.

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

## A  IMPLEMENTATION DETAILS

**Point Cloud Skeleton.** We apply voxel downsampling with a voxel size of 0.01m to the stacked city background point clouds, while the tracklet actors remain unchanged. In the depth map synthesis, when the number of tracked actor points falls below 20,000, we replace them with clustering template. During the computation, we only consider point clouds within a 100m radius of the ego vehicle. The projected high-resolution points map is pooling-downsampled to a size of 224×400.

**Generator.** For the nuPlan-mini dataset, we use a fixed text prompt: "a driving scene in {city}". Following existing protocols (Gao et al., 2023), for nuScenes, we directly use the original annotations and the model is trained with a learning rate of 1e-4, using cosine annealing with a 3k-step warm-up. During testing, we adopt UniPC with 20-step inference.

**Interactive Divergence Track Construction.** For the nuPlan-mini dataset, we select three scenes per scenario type from the training split, resulting in around 200 test scenarios. Each scenario starts from the ground truth of its first frame, and we generate right-lane change camera data for the ego vehicle. We apply a Gaussian blur with a kernel size of 3 to the point mask, making the resulting point cloud denser in the projected view. For IoU test: we assign a category label to each decoupled actor point when constructing the point cloud skeleton. During rasterized point-image projection, these point-level labels provide semantic ground truth. For evaluation, we use an official Mask2Former

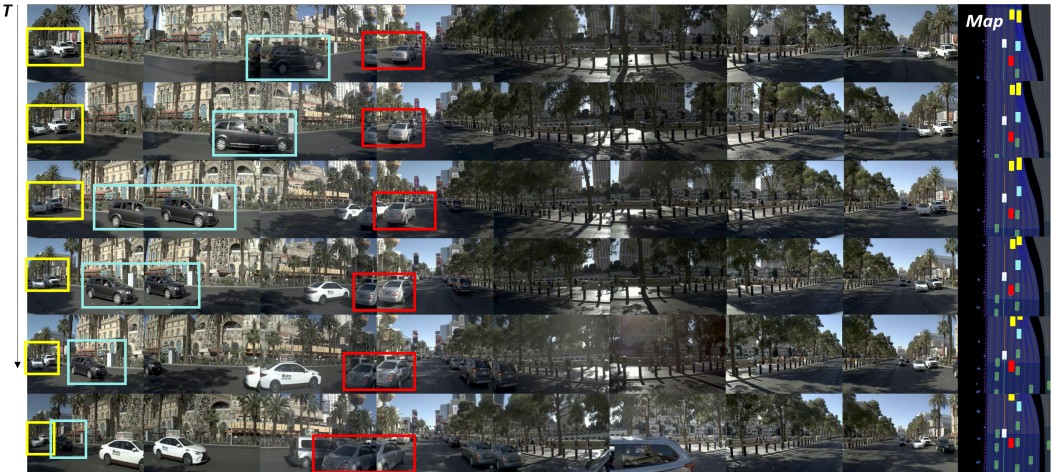

Figure 10: **Log Trajectory Ground Truth.** The ego car is closely following car in red bounding box.

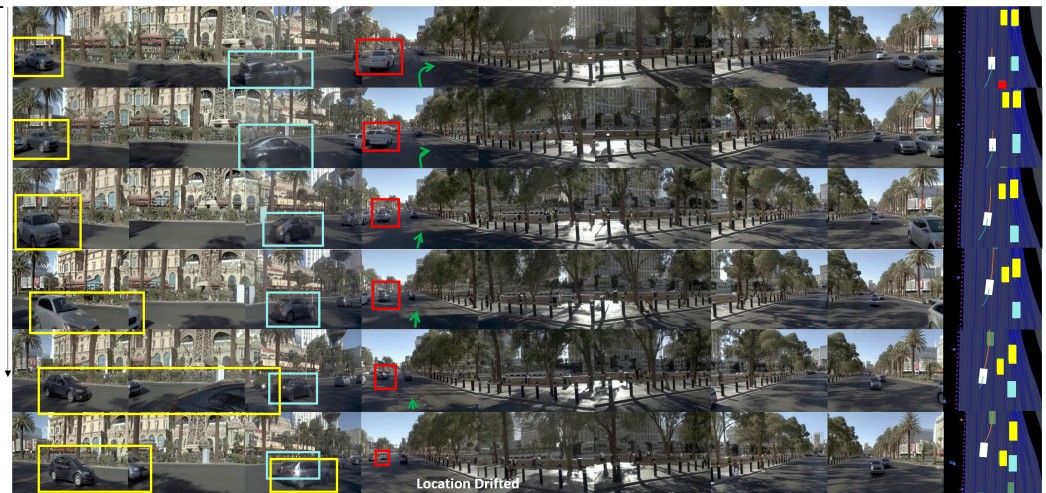

Figure 11: **Interactive Divergence Frames.** The ego car slows down and changes lanes, allowing the following vehicle to pass.

checkpoint trained on the Cityscapes Cordts et al. (2016) dataset to compute segmentation scores on Vehicles. This public dataset-evaluation setting ensuring the fairness.

Considering that lane-change maneuvers are typically completed within the first 80 frames, we define the test duration as 80 frames per scenario, generate images every 2 frames. The used split of nuPlan involve:

2021.05.12.22.28.35_veh35_00620_01164;
2021.05.12.22.00.38_veh35_01008_01518;
2021.05.12.23.36.44_veh35_02035_02387;
2021.05.12.23.36.44_veh35_01133_01535;
2021.05.12.23.36.44_veh35_00152_00504;

The specific test scenario ID will openaccess in code.

## B INTERACTIVE DIVERGENCE TRACK

We provide more detailed visual examples, and more visualization is in the video supplements. As shown in the Fig. 10 & Fig. 11. Notably, in the generated sequence controlled by the ego vehicle's trajectory-Fig. 11, other vehicles continue to follow their original paths, while the ego vehicle slows down and changes lanes. As a result, a vehicle that remained behind in the original trajectory overtakes the ego vehicle, forcing the model to generate a time-space-decoupled image sequence.

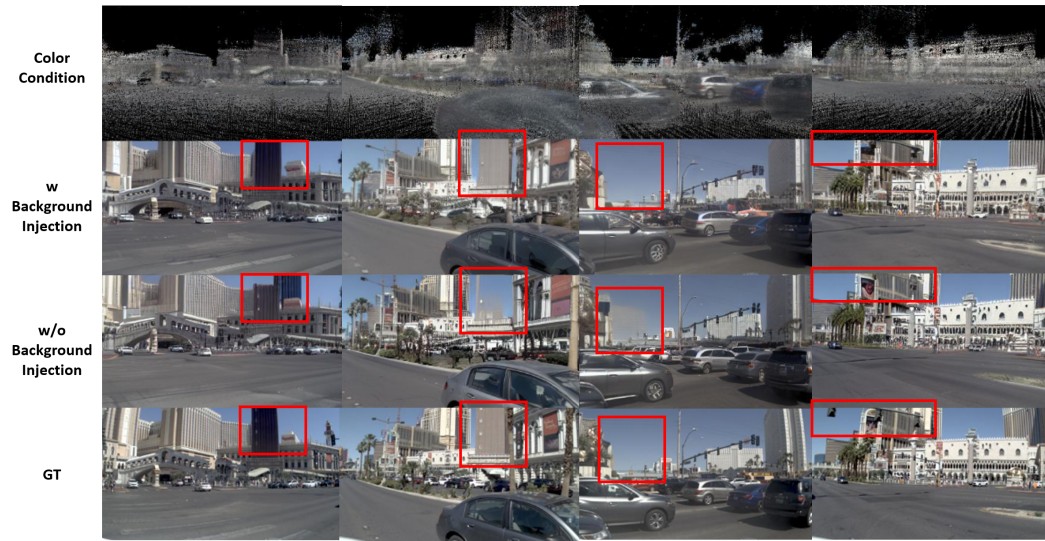

Figure 13: **Ablation of Background Injection.** Background Injection won't hugely affect the results but help recovery signals such as traffic boards.

## C  BACKGROUND INJECTION

Given the inherent limitations of point clouds in capturing upper-side scenery and sky regions, as shown in Fig. 12, background synthesis must rely on supplementary image cues. Since the previous condition need to be degraded by noise, we introduce a background branch to inject clear distant scenery for potential key traffic signals. Specifically, along the recorded trajectory, we anchor $N$ multi-view images at fixed distance. During both training and simulation, given the current ego-vehicle coordinates, we select two frames from neighbor anchor frames. Taking the ego vehicle's heading direction as the forward axis, the two selected frames are taken from the forward and backward directions, respectively. Subsequently, we crop the upper half of each image to remove most foreground vehicle pixels, **ensuring that vehicles from the recorded viewpoint do not interfere with actor generation along the simulated trajectory.** In the end, the anchor-to-ego camera transformations are embedded into the image features, which then interact with the conditional features through cross attention, as illustrated in Fig. 3.

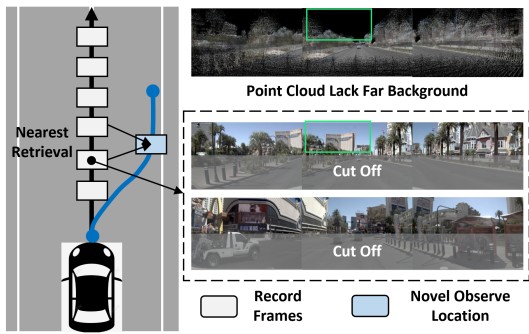

Figure 12: **Obtaining Far Distance Background.**

As ablation study illustrated in Fig. 13, when background injection is removed, the generative model performance won't decrease significantly, mostly in foreground actors. However, without background injection leads to loss of detail in specific background elements such as traffic boards. Considering most background information is not necessary in current E2E-AD models, we set this an optional module of *PointARSim*.

## D  SKELETON PROCESSING

**Skeleton processing.** Transforming and projecting point clouds across world/ego/camera frames is the main bottleneck. We therefore use a GPU projection operator at *inference* to accelerate this step. For *training*, we pre-project and cache the point images on disk.

**Efficiency vs. quality.** Table 5 reports the trade-off between voxel size and both generation quality and preprocessing cost. On nuPlan with 32 CPU workers, making the voxel grid overly fine brings negligible gains in $FID_I/FID_S$ but substantially increases cache time. At inference, the projection cost is largely insensitive to voxel size thanks to the GPU operator: projecting an *8-view* bundle at

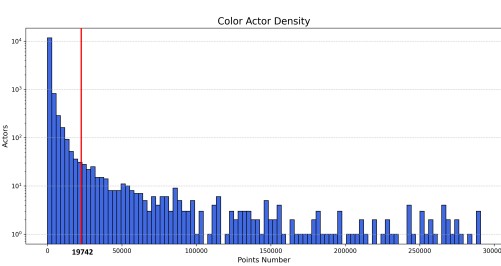

Figure 14: **Color Actor Points Number Distribution.**

Table 6: **Effect of previous condition noising step.** Same metric as Tab. 4, asynchronous noise adding obviously change the training objective, enhancing simulation.

| step mode | Synchronous | Asynchronous | | |
|---|---|---|---|---|
| Timestep | $\sim$ | [0-200] | [50-200] | [100-300] |
| FID (sim) | 29.35 | 27.86 | **27.73** | 27.98 |
| IoU (sim) | 55.71 | 60.43 | **61.31** | 59.02 |

$1080 \times 1920$ per view (then downsampled to $224 \times 400$ for the generator) takes **7.14 s** wall-clock on our hardware. The downstream image generation with UniPC (20 steps) takes about **20 s** per 8-view bundle, consistent with our chosen generative backbone. While our skeleton projection supports arbitrary output resolutions by adjusting the rasterization size and resize ratio, we keep the above resolution for fair comparison with prior work.

**Template replacement for sparse actors.** For depth-conditioned actors, any colorized actor with fewer than $N$ points is replaced by a clustered template. We set $N$ using the Jenks Natural Breaks algorithm over the per-actor point-count distribution. As shown in Fig. 14, **311** actors exceed this threshold and are kept, whereas **13,329** are replaced—indicating that only a small fraction of actors contribute raw depth during projection.

Table 5: **Voxel Size of Skeleton.** Besides generation performance, we also provide data preparing time in both training (CPU workers) and testing(GPU).

| Voxel Size | $FID_{I(\downarrow)}$ | $FVD_{I(\downarrow)}$ | $FID_{S(\downarrow)}$ | Seg IoU(%) | Cache(CPU/h) | Inference(GPU/s) |
|---|---|---|---|---|---|---|
| 0.1 | 10.15 | 112.52 | 28.92 | 54.18 | $\sim$4 | 6.83 |
| 0.05 | 9.02 | 102.52 | 27.87 | 60.42 | $\sim$12 | 6.93 |
| 0.01 | 8.87 | 100.34 | 27.73 | 61.31 | $\sim$24 | 7.14 |
| 0.005 | 8.64 | 100.13 | 27.62 | 61.56 | $\sim$36 | 8.35 |

## E  NOISE STEP SELECTION

**Noise step design.** In *PointARSim*, both the previous-condition design and the cycle loss rely on how the diffusion timestep $t$ controls image content. As shown in Fig. 15, for multi-view driving images, when $t < 200$ the global layout and color remain largely intact while high-frequency details (textures/edges) are attenuated. Using such *lightly noised* images as the previous condition gives stable style/structure cues without letting the model overfit to low-level patterns.

**Synchronous vs. asynchronous noise.** We evaluate different ways to inject noise into the previous condition (Tab. 6). With *synchronous* noise ($t' = t$), the model sees a previous latent $z_{t'}$ that is too similar to the current $z_t$ in driving scenes (high inter-frame redundancy), which creates a shortcut toward identity-like mappings and hurts simulation quality. With *asynchronous* noise, we decouple $t'$ from $t$. Including the clean boundary ($t' = 0$) lets non-degraded images leak in and weakens generative learning. While pushing $t'$ too high overly degrades temporal cues. A simple uniform range $t' \in [50, 200]$ strikes the best balance (FID **27.73**, IoU **61.31**) and is our default.

## F  STATIC VISUALIZATION

We provide visualization results for record trajectory on both the nuScenes and nuPlan datasets as shown in Fig. 16 and Fig. 16.

## G  USE OF LARGE LANGUAGE MODELS (LLMS)

During the preparation of this manuscript, we utilized a Large Language Model (LLM) to assist with improving the clarity, grammar, and readability of the text. Specifically, we used Google's Gemini model.

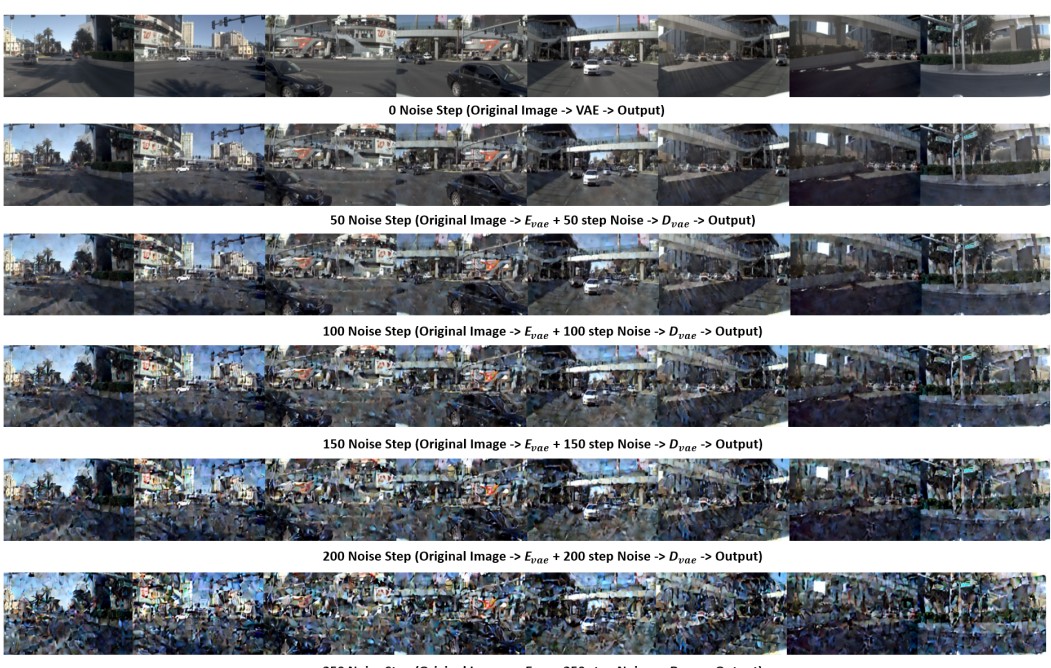

0 Noise Step (Original Image -> VAE -> Output)

50 Noise Step (Original Image -> $E_{vae}$ + 50 step Noise -> $D_{vae}$ -> Output)

100 Noise Step (Original Image -> $E_{vae}$ + 100 step Noise -> $D_{vae}$ -> Output)

150 Noise Step (Original Image -> $E_{vae}$ + 150 step Noise -> $D_{vae}$ -> Output)

200 Noise Step (Original Image -> $E_{vae}$ + 200 step Noise -> $D_{vae}$ -> Output)

250 Noise Step (Original Image -> $E_{vae}$ + 250 step Noise -> $D_{vae}$ -> Output)

Figure 15: **Multi-view Frame Noising Affect under UniPC-1000 Scheduler.**

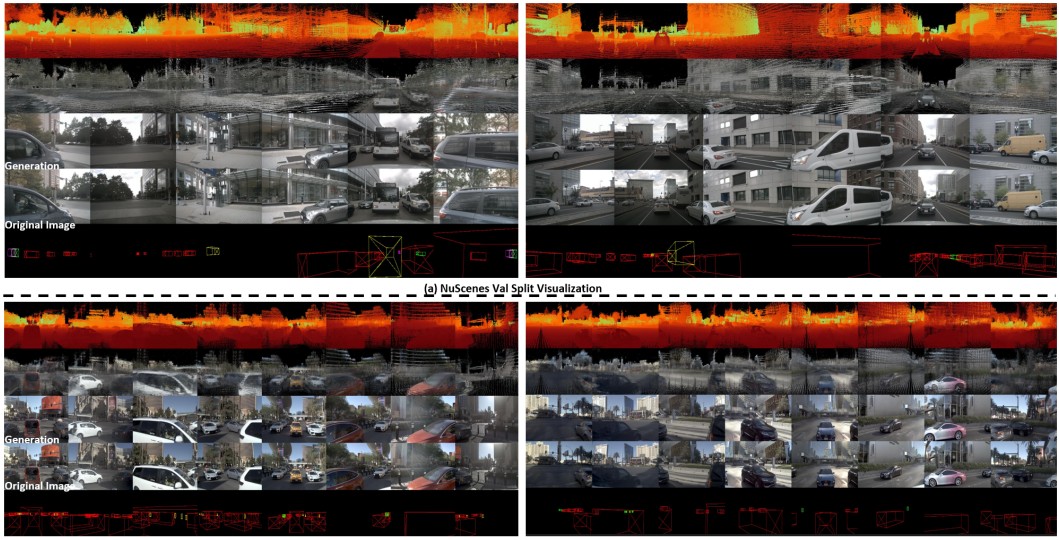

(a) NuScenes Val Split Visualization

(b) nuPlan Log-Replay Visualization

Figure 16: **Log Trajectory Visualization.**

The role of the LLM was limited to that of a writing assistant for tasks such as rephrasing sentences, correcting grammatical errors, and suggesting alternative phrasings. All the core ideas, scientific contributions, experimental design, and analysis presented in this paper are the original work of the authors. The LLM was not used for generating hypotheses, conducting experiments, or producing any of the results or conclusions of this research.

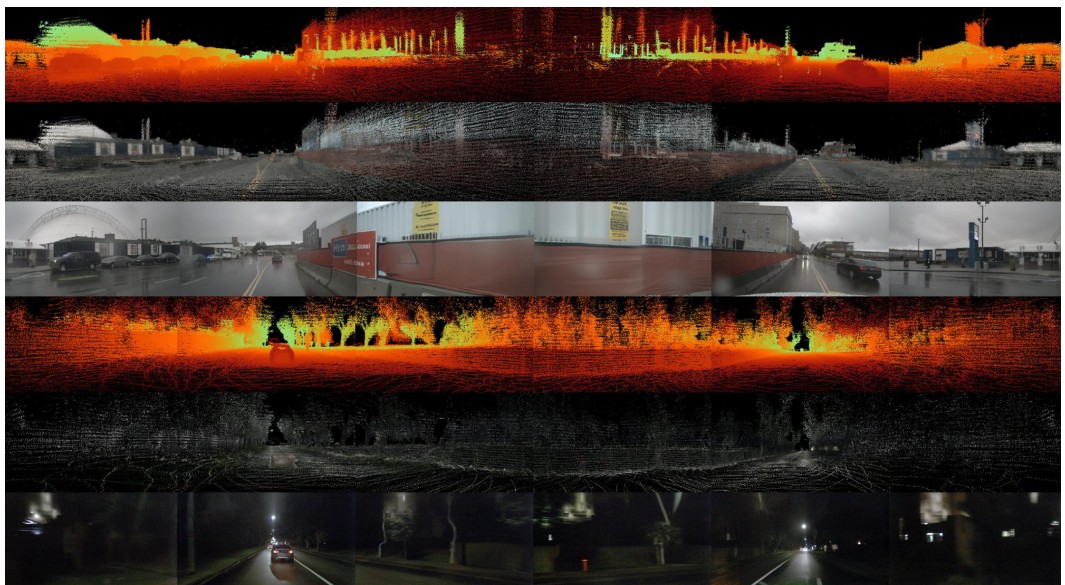

Figure 17: **Different Weather Visualization.**

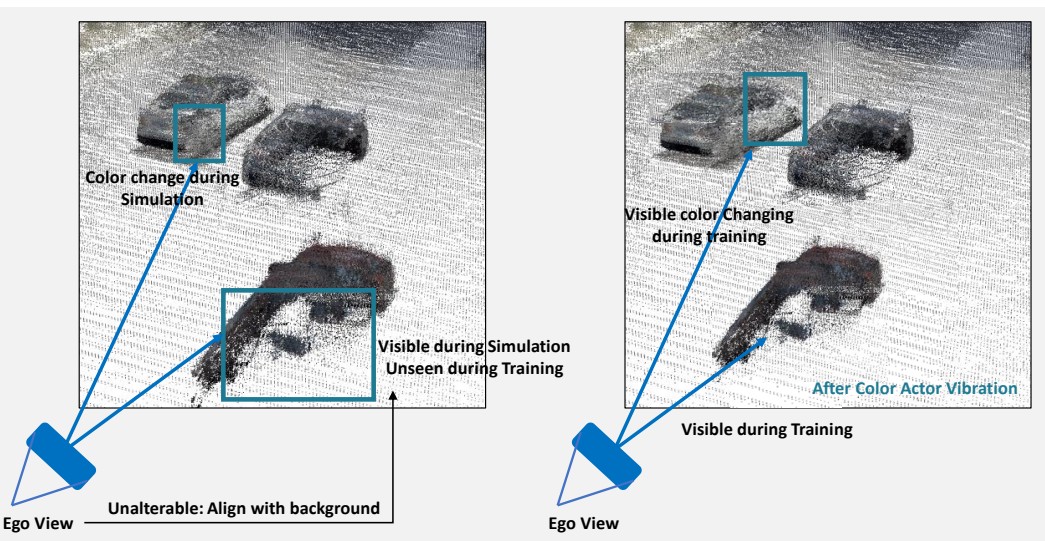

Figure 18: **Train-Val Gap Pattern.** We illustrate the train–validation gap pattern that CAV seeks to mitigate.

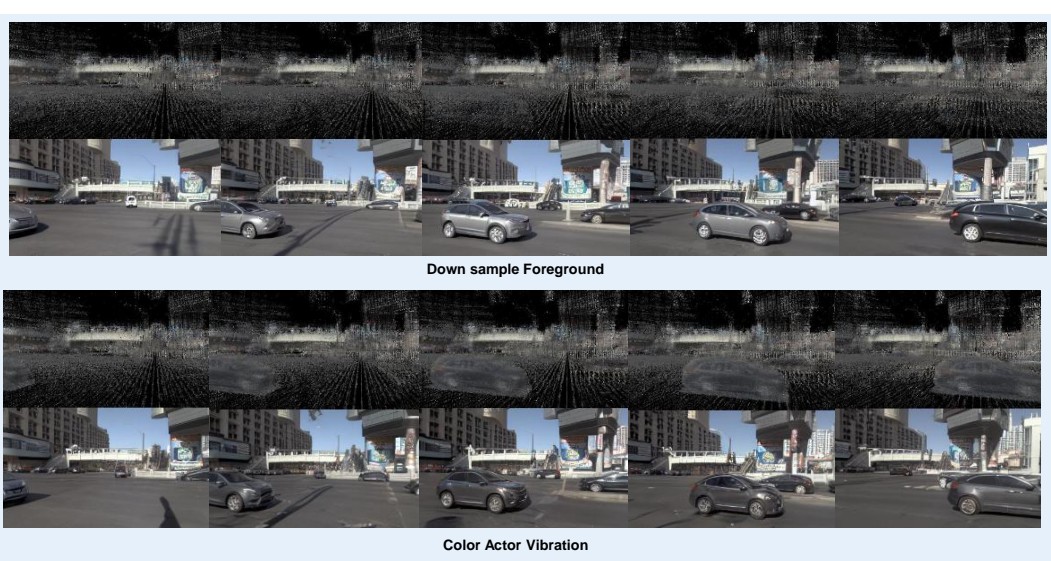

Figure 19: **Voxel DownSample Augmentation.** Foreground downsampling reduces the number of projected pixels, leading to decreased consistency.

