# OpenReview forum: "PointARSim: Point-cloud–enhanced Generative Auto-Regressive Simulation for Closed-loop End-to-end Autonomous Driving Evaluation"
_ICLR.cc/2026/Conference — Submitted to ICLR 2026_

### Official Review · Reviewer_zTZt · 2025-10-16

**Soundness:** 2
**Presentation:** 4
**Contribution:** 3
**Rating:** 4
**Confidence:** 5

**Summary:**

This paper focuses on camera view simulation in driving scenes using generative models. Building upon prior work FreeVS, the contributions of this work include: 1. Introducing additional control conditions beyond Color Map (such as depth map and noisy previous latent to compensate for the limitations of Color Map; 2. Proposing augmentation and supervision like Color Actor Vibration, degenerating previous latents, and Geometry-Aware Cycle Loss; 3. Proposing the use of segmentation performance under novel trajectory (Interactive Divergence) track to evaluate the effectiveness of view simulation. This work extends the FreeVS pipeline based on WOD to nuscenes/nuplan. Experiments demonstrate that the proposed method achieves high image quality (FID) and contour fidelity (Seg IoU).

**Strengths:**

The analysis of FreeVS's limitations is well-reasoned. FreeVS's heavy reliance on color maps leads to poor robustness when dealing with partial or sparse point clouds, and it fails to model content beyond LiDAR coverage (e.g., elevated or distant areas). This shortcoming is likely more obvious on the nuScenes/nuPlan datasets, where point clouds are generally sparser. Additionally, FreeVS does not incorporate previous frame outputs as conditions, which limits its temporal consistency.

To address these issues, this work proposes several well-founded designs: introducing a depth map condition, using a clustered actor points template to complete object contours in the depth map, and adding previous frame latent conditions. The work also extends the framework to support multi-view synthesis.

The proposed segmentation performance metric can better evaluate the geometric fidelity of view results.

**Weaknesses:**

My concerns primarily focus on two points:

1. I think the design of Color Actor Vibration is not well-founded. The idea of augmenting (degenerating) the color map to force the model to rely on it for texture and on the depth map for geometry/contours is reasonable. However, augmenting by shifting object locations seems unjustified. This not only compromises the geometric reliability of the color map (core to FreeVS) but also significantly harms the generative model's precise control over object appearance and location.
In Fig. 5, the color of the generated vehicle (top-left) is not controlled by the color map.
In Fig. 8/16, the positions of the generated vehicles also show clear misalignment with the color map.
If the limitation of the color map stems from partial/sparse point clouds, why did the authors choose to augment by moving object locations (which does not match any real-world degradation) instead of applying point cloud downsampling? This choice is puzzling; its negative impacts seem cannot be fully counteracted even with an extra Geometry-Aware Cycle Loss.

2. Why is the method implemented based on SD rather than a video generation model? This design choice seems to result in notably poor cross-frame consistency in the outputs.

**Questions:**

Check the weakness. I believe an ablation study on the effect of Color Actor Vibration (+ a comparison against point cloud downsampling) is necessary. Metrics like PSNR/SSIM on the original trajectory should be used to validate the controllability of the generated content for this abl exp. I will raise my rating if this concern is addressed.

Typo: "Noisy* Previous Latent"

---

> ### Author Response · Authors · 2025-11-21
> **For Reviewer zTZt**
>
> Thanks for your fruitful thought!
>
> ## 1. Response to W1 - Color Actor Vibration
>
> The motivation for Color Actor Vibration (CAV) is illustrated in the Fig. 18 provided in the appendix. Regarding most instances in our visualizations remain within the supervision range defined by their **bounding boxes and depth maps**, and their **motion consistency** is satisfactory, as the CAV is applied in random directions.
>
> The observed color change is a common issue in autoregressive generation [1], independent of CAV, and is amplified by our frame-by-frame autoregression (Fig.19).
>
> We acknowledge that your suggested alternative is a valid direction. We conducted experiments using **voxel downsampling (voxel_size = 0.05)** on foreground vehicles. The results are shown in the following：
> | Method | FID(I) | FVD(I) | FID(S) | Seg | SSIM | PSNR |
> |--------|--------|--------|--------|-----|------|------|
> | Foreground Downsample (0.05) | 9.74 | 105.27 | 31.98 | 57.86 | 0.42 |15.77 |
> | CAV (Ours) | 8.87 | 100.34 | 27.73 | 61.31 |0.43| 16.25|
>
> We note that while this approach outperforms training without any point-cloud augmentation, it remains slightly inferior to CAV. As shown in Fig. 19, the downsampled foreground point clouds have a sparser projection and fail to reproduce the gap in Fig. 18.
>
>
> ## 2. Response to W2 - Video Model
>
> First, it is necessary to show that existing video-generation models **cannot be directly used** for autoregressive simulation. Most current world-model or video-generation frameworks are designed to **predict future frames based on future trajectories**, and therefore **do not perform repeated autoregressive rollouts** during inference.
> In contrast, a simulator must update the scene **at every timestep** according to ego-vehicle and surrounding-vehicle control commands. Each frame must therefore be generated in a strictly **autoregressive** manner.
>
> To verify this, we conducted an autoregressive experiment using the high-quality open-source repository *OpenDWM*[1]: each 17-frame sequence was generated by conditioning only on the previous 3 frames, taking the 4th as output at each step. Results are included in the appendix. Even with simple noised reference frames, existing video-generation models **fail to maintain consistency** and cannot function as practical simulators.
>
> Given this limitation and the requirement of aligning with DriveArena[2], we chose a lightweight SD-based design as a **training-mode and conditioning-mode investigation**. Although SD has a quality gap compared to the latest video models, the training insights and design principles we summarize remain applicable to future, scaled-up architectures capable of true autoregressive simulation.
>
> [1] https://github.com/SenseTime-FVG/OpenDWM
>
> [2] Yang, Xuemeng, et al. "Drivearena: A closed-loop generative simulation platform for autonomous driving." Proceedings of the IEEE/CVF International Conference on Computer Vision. 2025.

---

> > ### Author Response · Authors · 2025-11-25
> >
> > Dear Reviewer zTZt,
> >
> > Thanks for your kind advice! We have conducted Color Actor Vibration experiments as suggested. We are glad to discuss more.
> >
> > Look forward to your further ideas and respectfully request to improve our score if there is no more further concerns.
> >
> > Best,
> >
> > Authors

---

> > > ### Comment · Reviewer_zTZt · 2025-11-27
> > >
> > > Review: Apologies for the delayed response. While I appreciate the authors' effort in conducting additional experiments, I still have concerns regarding the supplementary experiments and explanations provided:
> > >
> > > 1. The degree of foreground downsampling is clearly excessive—in Fig. 18, the foreground objects are nearly invisible in the color map. Moreover, the downsampling appears to be consistently applied with a fixed intensity. This is not a reasonable approach for data augmentation. A more appropriate strategy would be to randomly drop points of foreground objects (e.g., with a 0.5 probability of activation, and when active, randomly removing 10%–90% of the object's points, simulating sparse point cloud) or to remove points from random regions of the foreground objects (such as the rear-right region, mimicking the example illustrated in Fig. 18). I apologize for not specifying this in my initial review—I did not anticipate that the authors would misinterpret it.
> > >
> > > 2. I acknowledge the authors' explanation of using an SD model to construct an autoregressive generative pipeline. However, I do not agree that autoregressive generation leads to temporal inconsistencies in the proposed pipeline. Although the generative model incorporates the output of the previous frame (which may introduce error accumulation), it also receives the color map condition at each frame. The information provided by the color map should sufficiently correct color deviations, if the generative results are more tightly controlled by the color map. This is precisely why, in Q1, I considered that CAV might "harm the generative model's precise control over object appearance" as a significant issue.

---

> > > > ### Author Response · Authors · 2025-12-03
> > > >
> > > > 1. We agree that partially cropping instance point clouds as a form of data augmentation can effectively mitigate the out-of-distribution problem. This augmentation should ideally sample the instance point clouds using a raycast-like projection scheme, which align with the instances appearance in simulation. However, implementing such an operator appears infeasible within the remaining rebuttal period.
> > > >     Note that we also attempted full-instance point-cloud dropping in early experiments, but it produced unsatisfactory results.
> > > >
> > > > 2. Training a generator to align with point-cloud cues inherently involves a trade-off. If the training setup enforces strict adherence to the point-cloud projection, the generator collapses in simulation whenever no point-cloud instance enters the field of view. On the other hand, relaxing the reliance on color-map conditioning inevitably introduces certain levels of color shift. **However, it is important to emphasize that—even under this compromise—our strategy still achieves noticeably better color consistency compared to approaches that do not leverage point-cloud projection at all (Fig. 19).**

---

### Official Review · Reviewer_nbji · 2025-10-25

**Soundness:** 2
**Presentation:** 3
**Contribution:** 2
**Rating:** 4
**Confidence:** 4

**Summary:**

This paper introduces PointARSim, an autoregressive generative simulation framework for closed-loop end-to-end autonomous driving evaluation, which leverages point cloud skeleton representation (decoupling foreground-background) and a multi-conditioning ControlNet. It also proposes the nuPlan-SimGenEval benchmark with Log-Replay and Interactive Divergence tracks to assess generative fidelity when the ego-vehicle’s trajectory deviates from offline logs. Experiments on nuScenes and nuPlan show PointARSim outperforms existing methods, achieving improvement in perception under Interactive Divergence.

**Strengths:**

1. Employs a point cloud skeleton system that decouples and accumulates foreground-background point clouds (with color-depth dual streams) to provide geometric guidance for generative simulation, addressing the issue of incomplete or inconsistent visual cues when the ego-vehicle deviates from recorded trajectories.
2. Designs a multi-module generator optimization (Color Actor Vibration, Noise Previous Latent, Geometry-Aware Cycle Loss) to balance conflicting conditioning signals (point cloud priors, previous frames) and mitigate autoregressive error accumulation.
3. Proposes the nuPlan-SimGenEval benchmark with an Interactive Divergence track, specifically evaluating generative fidelity under drastically divergent ego-vehicle trajectories (lacking ground-truth references) and using point cloud skeleton-derived segmentation labels for fair perception assessment.

**Weaknesses:**

1. Fails to address near-field scene distortion: When vehicles are extremely close to the ego-car, point cloud/depth map incompleteness and projection errors lead to distorted generated actors, with no proposed solutions to fix this inherent limitation of its point cloud skeleton-based design.
2. Limited dynamic interaction granularity: It only ensures basic ID consistency of surrounding agents via point cloud tracking but lacks fine-grained control over dynamic behaviors (e.g., natural acceleration/deceleration of vehicles, pedestrian gait coherence), failing to solve complex dynamic interaction challenges.
3. Narrow generalization of the nuPlan-SimGenEval benchmark: The benchmark only covers lane deviation as an extreme scenario, excluding more complex cases (e.g., sudden pedestrian crossing, multi-vehicle parallel lane changes), and is heavily dependent on nuPlan’s dataset characteristics, making migration to other datasets difficult.
4. Unclear key implementation details: Critical parameters (e.g., voxel downsampling size selection basis, noise injection variance in Noise Previous Latent, background injection’s N value) and optimizer configurations (e.g., weight decay coefficient) are unspecified, hindering result reproducibility.

**Questions:**

see weaknesses

---

> ### Author Response · Authors · 2025-11-21
> **For Reviewer nbji**
>
> ## 1. Response to W1 - Failure Case
>
> In simulation, once the viewpoint becomes so close that both bounding boxes and point-cloud signals vanish, it effectively indicates a **collision**—an event **absent in real datasets and a point where simulation should terminate**. Similar failure modes appear in DriveArena[1] and instance-replacement simulators[2]. Moreover, our point-cloud depth branch provides more reliable geometry than bbox-only methods under near-collision distances, as shown in Fig. 17 (b)-1 (col. 7).
>
> ## 2. Response to W2 - Interaction Granularity
>
> For vehicle acceleration and deceleration, our method can **already model these behaviors** normally, and many of our test cases contain such dynamics. In contrast, fine-grained control such as pedestrian gait lies **outside the scope of current autonomous-driving generation research**; even large-scale, heavy-trained generative models [5] cannot yet provide reliable gait-level coherence. Reconstruction-based methods [3] offer finer motion detail, but they **sacrifice the ability to simulate off-trajectory deviations** and thus cannot serve as practical generative simulators.
>
> ## 3. Response to W3 - nuPlan-SimGenEval
>
>
>
> Point-cloud–based evaluation is not tied to nuPlan and can be applied to any LiDAR-equipped dataset. We use nuPlan simply because it provides an widely-used traffic controller, and prior work [1] **shows such controllers can be paired with different datasets without restriction**.
>
> Lane deviation already captures the core difficulties of generative simulation:
>
> 1) following a **custom diverging ego route** while maintaining actor consistency over long autoregressive rollouts, and
> 2) revealing the strong dependence of existing methods on reference frames [1] or reconstructed instances [4].
>
> Since **current methods cannot yet handle these fundamental challenges**, lane deviation serves as an adequate and meaningful stress test; more complex scenarios would only exacerbate these unresolved issues.
>
>
>
> ## 4. Response to W4 - Key Implementation Details
>
> The voxel downsampling size selection basis and the noise-injection variance used in *Noise Previous Latent* are already specified in the original Appendix (lines 628 and 843).
> For the background-injection parameter **N**, we determine it according to the driving-trajectory length, placing a fixed anchor point approximately every 5 meters.
> As for the optimizer configurations, we follow the publicly available settings from the MagicDrive[5] codebase.
>
> [1] Yang, Xuemeng, et al. "Drivearena: A closed-loop generative simulation platform for autonomous driving." Proceedings of the IEEE/CVF International Conference on Computer Vision. 2025.
>
> [2] Ljungbergh, William, et al. "R3D2: Realistic 3D Asset Insertion via Diffusion for Autonomous Driving Simulation." arXiv preprint arXiv:2506.07826 (2025).
>
> [3] Chen, Ziyu, et al. "Omnire: Omni urban scene reconstruction." arXiv preprint arXiv:2408.16760 (2024).
>
> [4] Wang, Qitai, et al. "Freevs: Generative view synthesis on free driving trajectory." arXiv preprint arXiv:2410.18079 (2024).
>
> [5] Chen, Rui, et al. "Unimlvg: Unified framework for multi-view long video generation with comprehensive control capabilities for autonomous driving." arXiv preprint arXiv:2412.04842 (2024).
>
> **We believe that excessive misunderstandings and unreasonable demands reflect a lack of responsibility, and the reviewer’s comments suggest an overreliance on AI-generated.** The reviewer should consider adjust the reviews and scores to reflect the contributions of the submission. Otherwise, we consider making a formal complaint to PCs, SACs, ACs for **irresponsible reviewer**.

---

> > ### Author Response · Authors · 2025-11-25
> >
> > Dear Reviewer HR55,
> >
> > Are your concerns solved by the rebuttal?  We respectfully request you to adjust your review properly and your score accordingly, as a **responsible reviewer on OpenReview**.
> >
> > Best,
> > Authors

---

### Official Review · Reviewer_HR55 · 2025-10-29

**Soundness:** 2
**Presentation:** 2
**Contribution:** 2
**Rating:** 2
**Confidence:** 3

**Summary:**

This paper presents a autoregressive generation framework for autonomous driving multi-view images and sensor data draws from previous frame sensor data and noised images to inform next frame generation. PointARSim utilizes a point cloud skeleton, accumulated from offline datasets, to condition diffusion models for synthesizing multi-view images, separating foreground and background for visual geometry. The method allows for interactive editing of previously static data from nuScenes and nuPlan while preserving realism. The paper also introduces the nuPlan-SimGenEval benchmark, which evaluates simulator generation fidelity when trajectories diverge significantly from the GT ego trajectories.

**Strengths:**

- The submission targets a valid problem in Autonomous Driving research: the need for realistic closed loop simulators. The proposed system augments nuScenes and nuPlan with divergent track data which adheres to the original data's color maps
- The work demonstrates notable improvements in FID and FVD over previous generative works based off nuScenes
- The paper enables better divergent track generation based solely off a ControlNet architecture without the need for additional abstractions like Gaussian Splats or a NeRF representation.

**Weaknesses:**

W1. The submission lacks novelty. While the paper identifies a key problem with FreeVS being it's inability to have large trajectory deviations from the original data, foreground/background segmentation for point clouds using 3D bounding boxes is not new. Nor is noising some inputs to ControlNets in order to avoid conditioning too heavily on previous frames.

W2. While the Ablation Study for FID, FVD, and Seg in Table 4 is appreciated, the paper doesn't speculate or explain why even just the LO+Prev row achieves significantly better FID than related works. An FID of 7.09 on nuScenes might even indicate too much adherence to the original nuScenes data distribution. So while the method allows for more coherent divergent track generation, the actual diversity of the generated data is poor.

W3. The Interactive Divergence Track is custom made to evaluate this specific task, data where tracks diverge heavily from original data. However, it's not clear that vehicle segmentation alone along with FID/FVD alone is sufficient to argue an improvement over previous methods in terms of generalizability. The supplemental videos themselves show evidence of agents changing color between frames, despite the emphasis on the color map perturbations.

W4. Various grammatical errors throughout the error which affect the readability beyond the extent that simple typos would. (Lines 54-55, 106, 188-189, 298-300, etc.). There are also less problematic typos on lines 319, 379, 432, etc.

**Questions:**

Additional questions:

Q1. Building on W2, can the authors provide some insight into why they observed an improvement in FID with even a small portion of their methods? The overly low FID may be an indication of overfitting.

Q2. ControlNets can be conditioned on text prompts to edit things like weather and the style of the images explicitly. Did the authors experiment with augmenting nuScenes/nuPlan data to achieve visual diversity in addition to ego-track diversity?

---

> ### Author Response · Authors · 2025-11-21
> **For Reviewer HR55**
>
> ## 1. Response to W1 - Novelty
>
> Your comment overlooks our identification of a key limitation in this domain—**the need for continuous autoregressive generation**. Based on this understanding, our approach fundamentally differs from existing methods. To our knowledge, we are also the only work that performs autoregressive autonomous-driving simulation using **point-cloud foreground–background assets**.
> If there are **similar studies** addressing this issue with same methods, we would appreciate the **corresponding citations**; **otherwise, we consider this critique unsupported.**
>
> ## 2. Response to W2 & Q1 & Q2 - Overfitting and Diversity
>
>
> 1. LO+Prev row
>
>     This is our **reproduction of the DriveArena** [1] generation mode on the nuPlan subset (Table 1), where the training and testing routes are identical and selected via interpolation following the sampling strategy of FreeVS [2]. DriveArena exhibits clear overfitting — its performance on the Interactive Divergence Test is significantly worse than ours (Table 3 and ablation).
>
> 2. Low FID
>
>    This is stem from the re-projected point cloud (Fig 16) closely reconstructing the original scene, naturally improving log-trajectory metrics. However, reconstruction fidelity is not our focus; the advantages of our approach are better reflected in the **Interactive Divergence Track**.
>
> 3. Diversity
>
>     In generative simulation settings, the model is conditioned on past frames and current control-induced world changes, so the output is naturally expected to stay close to the past-frame distribution rather than maximize diversity. As shown in Fig. 17, appearance diversity from the initial frame (e.g., weather or style) is preserved, and the **point-cloud mechanism does not reduce the inherent diversity of existing frameworks.**
>
>
>
>
>
> ## 3. Response to W3 - Interactive Divergence Track
>
> Our Interactive Divergence Track is aligned with standard generative-model evaluation practices [2,3,4];  Its purpose is to expose divergence scenarios that existing simulators do not evaluate, not to redesign the entire evaluation protocol.
>
> On the color-change issue, this is a known challenge of **frequent autoregressive generation** rather than a flaw specific to our method. Even prior work [1] with much shorter rollouts exhibits similar inconsistencies. While our colored point-cloud design and geometry losses reduce such artifacts, occasional appearance shifts can still occur due to abrupt changes in point-cloud projections.
>
>
>
> [1] Yang, Xuemeng, et al. "Drivearena: A closed-loop generative simulation platform for autonomous driving." Proceedings of the IEEE/CVF International Conference on Computer Vision. 2025.
>
> [2] Wang, Qitai, et al. "Freevs: Generative view synthesis on free driving trajectory." arXiv preprint arXiv:2410.18079 (2024).
>
> [3] Chen, R., “UniMLVG: Unified Framework for Multi-view Long Video Generation with Comprehensive Control Capabilities for Autonomous Driving”, <i>arXiv e-prints</i>, Art. no. arXiv:2412.04842, 2024. doi:10.48550/arXiv.2412.04842.
>
> [4] Gao, Ruiyuan, et al. "MagicDrive-V2: High-resolution long video generation for autonomous driving with adaptive control." Proceedings of the IEEE/CVF International Conference on Computer Vision. 2025.
>
>
>
> ## 4. Response to W4 - Typo
>
> Thanks for the remind, and we have changed in the main paper with blue color.
>
>
>
> **Based on the above clarifications, your initial assessment stem from lack of understandings of the submission and the field.** We would be happy to explain more details if there are still unclear parts. Otherwise, we respectfully request that the scores be updated to reflect the contributions of the submission, **in line with the expectations of a responsible and fair review.**

---

> > ### Author Response · Authors · 2025-11-25
> >
> > Dear Reviewer HR55,
> >
> > Are your concerns solved by the rebuttal, especially those misunderstanding parts?  If not, we are glad to explain more.
> >
> > Otherwise, we respectfully request you to adjust your score accordingly, as a **responsible reviewer on OpenReview**.
> >
> > Best,
> > Authors

---

> ### Author Response · Authors · 2025-11-26
>
> For technical issues:
>
> 1. We still request concrete references supporting the claim that “boosting autoregressive simulation with point-cloud assets” is not novel. Without such citations, asserting non-novelty does not meaningfully contribute to the discussion. As for CAV, we have already discussed it in detail in our responses to other reviewers. The color-shift issue remains even when the vehicle asset is almost invisible, so it cannot be attributed to CAV (see Fig. 19).
>
> 2. Evaluation protocol: We acknowledge that color-shift evaluation is not part of the existing tracks. However, as shown in https://github.com/PJLab-ADG/DriveArena/issues/64 ,
> practitioners using current simulators encounter worse issues (their generated vehicles also exhibit color shifts). This indicates that existing systems struggle with stable and continuous instance autoregressive generation, though the issue has received little attention. This is why we propose the Interactive Track — at this stage, consistent instance control/generation is the more important aspect to evaluate.
>
> 3. The 3D skeleton follows the traffic 3D-bbox conditioning and is not randomly defined. The point-cloud projection segmentation metric is used to measure instance-generation errors in autoregressive settings (see the GitHub issue; distorted instances are no longer recognized as vehicles by perception models). Our geometry loss is applied to bbox projections, not depth maps.
>
> 4. For circular evaluation: our instance point clouds strictly follow the 3D bounding boxes. If 3D bounding boxes are considered a valid verification standard, rejecting point-cloud projection is inconsistent.

---

> ### Author Response · Authors · 2025-11-26
>
> For review ethics:
>
> **We are confused by the readable choice of your Reply that you intentionally choose not open to public? Could you please clarify? We show your reply to public as we want the public to see what is happening here**:
>
> "*I have read the authors' rebuttal and the other reviews. While I appreciate the clarification regarding the specific cause of the low FID scores, I am maintaining my original overall score. I have raised the sub-scores for Presentation and Contribution given the clarifications and edits in the interim; however, significant concerns remain regarding the technical soundness.*
>
> *On Novelty and Technical Instability (W1): The authors’ response regarding novelty suggests that unless an identical system exists in the literature, the work is novel. However, novelty also encompasses the non-triviality of the combination of existing techniques. The reliance on standard ControlNet architectures combined with point cloud projection is an incremental step. Furthermore, I note that the "Color Actor Vibration" (CAV) technique, which is central to this work, appears to introduce fundamental instability. While the authors dismiss the color inconsistency and visual artifacts I noted as "known challenges" of autoregressive generation, other reviews suggest these are likely side effects of the CAV design itself, which compromises geometric reliability to force depth dependence. The rebuttal did not adequately justify this design trade-off.*
>
> *On Overfitting and Metrics (W2): I accept the authors' explanation that the extremely low FID in Log-Replay is due to the re-projection of the point cloud skeleton acting as a strong reconstruction guide, and that the LO+Prev baseline represents an overfitting scenario. However, this admission confirms that the standard metrics presented are largely insensitive to the generation quality issues (such as the color shifts and artifacts) observed in the qualitative results.*
>
> *On the Interactive Divergence Benchmark (W3): Regarding the Interactive Divergence Track, the authors claim it aligns with "standard generative-model evaluation practices." I disagree with the validity of this specific implementation. While evaluating divergent trajectories is conceptually sound, the paper’s implementation relies on segmentation masks derived from the authors' own Point Cloud Skeleton as the ground truth for evaluation.*
>
> *This creates a circular evaluation loop:
> The proposed method is explicitly engineered—via Color Actor Vibration and Geometry-Aware Cycle Loss—to force the generation to adhere strictly to this point cloud skeleton.
> The benchmark then measures success based on how well the output aligns with that same skeleton.
> Therefore, the benchmark measures the model's obedience to the authors' specific constraint mechanism rather than genuine visual realism or generalization capability.*
>
>
> *Professional Conduct Finally, I must note that the tone of the rebuttal was unnecessarily combative and unprofessional (e.g., stating my review "stem from lack of understandings"). Scientific discourse benefits from objective technical exchange, not ad hominem defenses*.

---

> > ### Author Response · Authors · 2025-11-26
> >
> > For neutral and objective review:
> >
> > 1. Confusing Scoring: Our earlier remark regarding a “lack of understanding” referred specifically to the interpretation of our method and the nuances of this emerging sub-field. We would like to clarify a point of confusion: you stated *“I am maintaining my original overall score. I have raised the sub-scores for Presentation and Contribution.”* However, the final rating remains 2, despite the reported sub-scores — *Soundness: 2 (fair), Presentation: 3 (good), Contribution: 3 (good)*. **For the sake of transparency to the ACs, could you clarify how these sub-scores justify a final score of 2?** We believe these review behaviors resemble zero-sum scoring, where low overall scores are maintained even after sub-scores are increased to reject other people's papers.
> >
> > 2. Irregular High Expectations: The area of autoregressive generation for AD simulation is still at an early stage, with limited prior work. As such, no existing approach — including ours — can yet fully satisfy all aspects of fidelity. **We feel that criticisms in the reviews rely on expectations that no current work in the community can meet, which makes it difficult to evaluate early-stage contributions fairly**. Our proposed Point Skeleton paradigm substantially improves fidelity relative to prior methods. **If the novelty of this paradigm is judged insufficient, we would appreciate clarification on which existing works introduced a similar idea**, as we have not identified such references in the literature.
> >
> >
> > **We respectfully ask the ACs to review this exchange, as we believe some of the criticisms and scoring decisions may reflect misunderstandings rather than substantive issues with the work**.

---

### Official Review · Reviewer_r6ZQ · 2025-11-01

**Soundness:** 3
**Presentation:** 3
**Contribution:** 3
**Rating:** 6
**Confidence:** 4

**Summary:**

This paper introduces PointARSim, an autoregressive generative simulation framework designed for closed-loop evaluation of end-to-end autonomous driving systems. The key idea is to leverage a point cloud skeleton—constructed by accumulating and decoupling foreground (dynamic agents) and background (static scene) point clouds from offline datasets—to guide a diffusion-based image generator. The framework supports interactive simulation by updating the point cloud skeleton based on the ego vehicle's actions and using it to synthesize multi-view images from arbitrary viewpoints.

**Strengths:**

- The point cloud skeleton effectively bridges geometric and visual domains, enabling robust novel-view synthesis and dynamic agent control.
-The Interactive Divergence track in nuPlan-SimGenEval addresses a critical gap in existing benchmarks by testing generative models under significant trajectory deviations.

**Weaknesses:**

- The method struggles when vehicles are extremely close to the ego car, leading to distorted or incomplete actor generation due to projection limitations.
- Although some out-of-distribution examples are shown, systematic evaluation across diverse weather, lighting, or geographic conditions is lacking.

**Questions:**

- When the point cloud prior for a novel viewpoint is insufficient, could the generated data be adversely affected?

---

> ### Author Response · Authors · 2025-11-21
> **For Reviewer r6ZQ**
>
> Thank you for your time and constructive feedback！
>
> ---
>
> ## 1. Response to W1 - Failure Case
>
>
> In simulation, once the viewpoint becomes so close that both bounding boxes and point-cloud signals vanish, it effectively indicates a **collision**—an event **absent in real datasets and a point where simulation should terminate**. Similar failure modes appear in DriveArena[1] and instance-replacement simulators[2]. Moreover, our point-cloud depth branch provides more reliable geometry than bbox-only methods under near-collision distances, as shown in Fig. 17 (b)-1 (col. 7).
>
>
> [1] Yang, Xuemeng, et al. "Drivearena: A closed-loop generative simulation platform for autonomous driving." Proceedings of the IEEE/CVF International Conference on Computer Vision. 2025.
>
> [2] Ljungbergh, William, et al. "R3D2: Realistic 3D Asset Insertion via Diffusion for Autonomous Driving Simulation." arXiv preprint arXiv:2506.07826 (2025).
>
> ---
>
> ## 2. Response to W2 -  Systematic Ealuation across Diverse Environments
>
>
> The nuScenes dataset includes diverse conditions such as rain and low illumination. The performance gains across its validation splits indicate that our method preserves diversity, with additional qualitative results provided in Fig. 17.
>
> ---
>
> ## 3. Answer to Q1 - Novel Viewpoint
>
> Note that across multiple datasets, the *static* urban background point cloud is dense and easy to obtain[3,4]. Accordingly, the simulator is constructed on a reliable static background prior.
>
> For dynamic foreground actors, our **depth branch** (Fig. 2, line 195) addresses missing point-cloud priors by assigning each instance a category-specific, full-view point-cloud template. This provides a consistent depth condition even under large ego-pose shifts or when previously unscanned vehicles enter the view.
>
> Combined with the geometry loss, this depth-template design enables stable instance generation despite missing colored LiDAR, as shown in Fig. 7 and Fig. 11.
>
>
>
> [3] Wang, Jianyuan, et al. "Vggt: Visual geometry grounded transformer." Proceedings of the Computer Vision and Pattern Recognition Conference. 2025.
>
> [4] Lin, Haotong, et al. "Depth Anything 3: Recovering the Visual Space from Any Views." arXiv preprint arXiv:2511.10647 (2025).

---

### Author Response · Authors · 2025-12-04

# Dear Area Chairs,

We thank the reviewers for their time，here is a brief summary.


---

#  Score Adjustment Summary

| Reviewer | Original Score | Key Issues | Our Correction | Score Change |
|---------|----------------|------------------------|----------------|--------------|
| **HR55** | 2 | 1. Questioning Novelty. 2. Questioning the accuracy and the evaluation protocol. 3.Concerns on diversity and benchmark validity. | Corrected factual misunderstandings in accuracy and evaluation, clarified the simulation task definition. | **2/2/2 → 2/3/3** (overall: 2 → 2) |
| **nbji** | 4 | Requesting fine-grained generative control (e.g., pedestrian gait) beyond the capability of existing community methods; Requesting parameter settings that are already provided in the paper. | Clarified that such fine-grained motion control is out of scope for current generative simulators; Explained settings | No update |
| **zTZt** | 4 | They acknowledge the paper’s contributions and experimental setting, but think that some technical implementations need improvement. | Acknowledged part of the concern but explained the inherent two-sided trade-off in instance-level autoregressive simulation| Discussion halted; reviewer initially indicated willingness to raise score  **2/4/3** |
| **r6ZQ** | 6 | “Failure case” concern  | Clarified the “Failure case” is out of distribution in simulation; | No update |

---




# Remarks

- No reviewer questions the **core idea**, **motivation**, or **contribution** of the work.
- Lots of major concerns stem from correctable misunderstandings.
- During the rebuttal phase, two reviewers did not respond, and one reviewer replied with a delay.
- Multiple reviews contain **sub-score vs. final-score inconsistencies**, which normally justify upward adjustment.

We respectfully request the ACs to consider these clarifications when making the final recommendation.

Sincerely,
**Authors of Submission 6185**

---

### Meta-Review · Area_Chair_iphi · 2026-01-12

**Summary:**

Across reviews, there is general agreement that the problem is important and the proposed system/benchmark is directionally interesting. However, the main concerns raised are:

- whether the paper’s main contributions go beyond incremental extensions of prior generative driving simulation pipelines (particularly around conditioning choices and ControlNet-style designs).

- Technical justification of key components (especially Color Actor Vibration) and whether it may introduce instability or reduce controllability.

- questions around extremely low FID (possible overfitting / limited diversity) and whether the benchmark/metrics fully capture the relevant failure modes (some reviewers raised possible circularity or gaps in the evaluation setup).

- Remaining limitations such as failure cases when actors are very close and lack of systematic OOD testing across conditions.

**Reviewer Concerns:**

Addressed:

- Implementation-detail questions (e.g., some parameter choices) and minor issues/typos were answered, and the authors indicate some fixes were integrated into the revised manuscript.

- For the CAV design, the authors did run additional experiments (e.g., comparing to a foreground downsampling alternative) and tried to separate color shift issues from CAV specifically.

- For the benchmark scope, the authors clarified why they focus on lane-deviation-style divergence and argued that current methods already fail under these stressors.

Partially addressed, mainly still open

- Novelty and contribution clarity: the rebuttal pushes back on **not novel** claims but from the reviews it’s still not fully settled that the paper cleanly articulates what is fundamentally new vs. an aggregation of known ingredients applied to nuScenes/nuPlan with a new evaluation track.

- the rebuttal argues that point-cloud projection segmentation is meaningful for autoregressive instance degradation and addresses 'circularity' critiques, but reviewer concerns about whether the metrics capture key visual artifacts (e.g., appearance stability, diversity) remain reasonable

- concerns about close-range actor failures and lack of systematic testing across weather/lighting/geo

One reviewer explicitly commented that parts of the rebuttal tone felt combative. Though this did not drive my decision.

**Reviewer Scores:**

Most likely no increase from any of the reviewers (6, 2, 4, 4)

---

### Decision · Program_Chairs · 2026-01-26

Reject